Common neural mechanisms supporting time judgements in humans and monkeys

Rodriguez-Larios Julio Julio.RodriguezLarios@brunel.ac.uk 1
Rassi Elie 2 3
Mendoza German 4
Merchant Hugo 4
Haegens Saskia 5 6
1 Life Sciences, Brunel University , London , United Kingdom
2 Department of Psychology, Centre for Cognitive Neuroscience, Paris-Lodron-University of Salzburg , Salzburg , Austria
3 Donders Institute for Brain, Cognition and Behaviour, Radboud University Nijmegen , Nijmegen , Netherlands
4 Instituto de Neurobiología, UNAM , Queretaro , Mexico
5 Department of Psychiatry, Columbia University , New York , United States of America
6 Division of Systems Neuroscience, New York State Psychiatric Institute , New York , NY , United States of America
Gollo Leonardo
Electronic publication date: 2024 Nov 19
Publication date: 2024
Volume: 12
Electronic Location ID: e18477
Received 2024 Jun 17; Accepted 2024 Oct 16
Copyright: ©2024 Rodriguez-Larios et al.
Copyright year: 2024
Copyright holder: Rodriguez-Larios et al.
License: This is an open access article distributed under the terms of the Creative Commons Attribution License, which permits unrestricted use, distribution, reproduction and adaptation in any medium and for any purpose provided that it is properly attributed. For attribution, the original author(s), title, publication source (PeerJ) and either DOI or URL of the article must be cited.
License URL: https://creativecommons.org/licenses/by/4.0/

Common neural mechanisms supporting time judgements in humans and monkeys 28 4 2024 2024.04.25.591075 bioRxiv 10.1101/2024.04.25.591075 PMC11071527 38712259
Keywords: Time perception, EEG, ERP, Non-human primates

Funding: The Austrian Science Fund (FWF) Erwin Schrödinger Fellowship J4580 UNAM-DGAPA-PAPIIT IG200424 and UNAM-DGAPA-PASPA NWO Vidi 016.Vidi.185.137 and NIH R01 MH123679 UNAM-DGAPA-PAPIIT IA202024 Elie Rassi is supported by the Austrian Science Fund (FWF) Erwin Schrödinger Fellowship J4580. Hugo Merchant is supported by UNAM-DGAPA-PAPIIT IG200424 and UNAM-DGAPA-PASPA. Saskia Haegens is supported by NWO Vidi 016.Vidi.185.137 and NIH R01 MH123679. German Mendozais supported by UNAM-DGAPA-PAPIIT IA202024. The funders had no role in study design, data collection and analysis, decision to publish, or preparation of the manuscript.

==============================
There has been an increasing interest in identifying the biological underpinnings of human time perception, for which purpose research in non-human primates (NHP) is common. Although previous work, based on behaviour, suggests that similar mechanisms support time perception across species, the neural correlates of time estimation in humans and NHP have not been directly compared. In this study, we assess whether brain evoked responses during a time categorization task are similar across species. Specifically, we assess putative differences in post-interval evoked potentials as a function of perceived duration in human EEG (N = 24) and local field potential (LFP) and spike recordings in pre-supplementary motor area (pre-SMA) of one monkey. Event-related potentials (ERPs) differed significantly after the presentation of the temporal interval between “short” and “long” perceived durations in both species, even when the objective duration of the stimuli was the same. Interestingly, the polarity of the reported ERPs was reversed for incorrect trials (i.e., the ERP of a “long” stimulus looked like the ERP of a “short” stimulus when a time categorization error was made). Hence, our results show that post-interval potentials reflect the perceived (rather than the objective) duration of the presented time interval in both NHP and humans. In addition, firing rates in monkey’s pre-SMA also differed significantly between short and long perceived durations and were reversed in incorrect trials. Together, our results show that common neural mechanisms support time categorization in NHP and humans, thereby suggesting that NHP are a good model for investigating human time perception.

Introduction

Time estimation in the range of hundreds of milliseconds is a crucial ability for many species, as it is necessary for a wide variety of behaviours including foraging and communication. The last decade has seen an increasing interest in the identification of neural underpinnings of motor and perceptual timing (Balasubramaniam et al., 2021; Tsao et al., 2022). Neurophysiological experiments have suggested the existence of a core timing network that includes the medial premotor areas (SMA and preSMA) and its (sub) cortical connections (Merchant, Harrington & Meck, 2013). The current hypothesis is that the medial premotor cortex encodes both elapsed time and the temporal scaling of its neural population trajectories in state space (Gámez et al., 2019; Sohn et al., 2019). These dynamics are linked to the existence of neural sequences that form patterns of active neurons changing in rapid succession and that flexibly cover an interval depending on the timed duration (Crowe et al., 2014; Merchant et al., 2015).

Electroencephalography (EEG) is an ideal tool to investigate the neural correlates of time estimation in humans due to its high temporal resolution. For this purpose, previous studies have combined EEG with different interval timing tasks (Bueno & Cravo, 2021; Damsma, Schlichting & Van Rijn, 2021; Duzcu, 2019; Lindbergh & Kieffaber, 2013; Ng, Tobin & Penney, 2011; Ofir & Landau, 2022; Özoğlu & Thomaschke, 2023; Pfeuty, Ragot & Pouthas, 2005). Most of these tasks involve the presentation of visual or auditory stimuli to signal a to-be-timed interval, which has to be compared to a reference time interval or prototype. Early studies focused on ERPs during the presentation of the time interval (Macar & Vidal, 2003; Pfeuty, Ragot & Pouthas, 2005). However, it has been recently shown that time estimation is better reflected in post-interval potentials (i.e., evoked responses emerging after the offset of the time interval) around fronto-central electrodes (Kononowicz & Penney, 2016; Kononowicz & Rijn, 2014). Specifically, when the presented stimulus is perceived as longer than the reference, a more pronounced positive potential around 200 ms (P200) has been found, while for shorter stimuli a more pronounced Late Positive Potential (LPP) and P300 have been found (Damsma, Schlichting & Van Rijn, 2021; Kononowicz & Rijn, 2014; Kruijne, Olivers & Van Rijn, 2021; Lindbergh & Kieffaber, 2013; Özoğlu & Thomaschke, 2023; Tarantino et al., 2010).

Although research in NHP has provided insights into the neural correlates of time estimation (Merchant, Harrington & Meck, 2013), these have not yet been directly compared to electrophysiological findings in humans. This is primarily because research in NHP is often focused on spiking activity (Leon & Shadlen, 2003; Mendoza et al., 2018), which cannot be recorded non-invasively in healthy human participants. Although no previous study has investigated post-interval potentials during time estimation tasks in monkeys, they are expected to be qualitatively similar to those found in humans. Indeed, the behaviour of monkeys and humans on time estimation tasks shows similar psychometric properties, which suggests a common neural substrate (Mendez et al., 2011; Zarco et al., 2009). Moreover, evoked potentials have been reported in monkeys in other cognitive tasks, showing similar dynamics to the ones observed in humans (Godlove et al., 2011; Peissig et al., 2007).

Here, we assess whether the neurophysiological signatures of time perception are similar in humans and NHP. For this purpose, we analysed EEG data from 24 humans and extracellular recordings in pre-SMA from one monkey while they performed a temporal interval categorization task. In this task, participants had to decide whether a visually presented temporal interval had a shorter or longer duration than a previously learned prototype. Based on prior work, we hypothesized that post-interval evoked potentials would significantly differ between “short” and “long” perceived durations in both humans and NHP, and that this would be accompanied by changes in firing rates in the latter.

Methods

Portions of this text were previously published as part of a preprint (see https://doi.org/10.1101/2024.04.25.591075).

Participants

Humans. 27 healthy adult subjects (12 males) participated in the experiment. The mean age was 25.6 years old (SD = 4.2). Participants reported normal or corrected-to-normal vision and no history of neurologic or psychiatric diagnosis. Informed consent procedure and study design were approved by the Institutional Review Board (IRB) of the New York State Psychiatric Institute (protocol #8001). Participants were compensated for their participation (at 25 USD per hour). Three participants were excluded from the analysis due to technical problems during data acquisition.

Monkey. Although the original study included two animals (Mendoza et al., 2018), we here only analysed the data of Monkey 1 because evoked responses in Monkey 2 could not be obtained due to a lower signal-to-noise ratio in the LFP recordings (see Fig. S1). All experimental procedures were approved by the National University of Mexico Institutional Animal Care and Use Committee and conformed to the principles outlined in the Guide for Care and Use of Laboratory Animals (NIH, publication number 85–23, revised 1985). The monkey was obtained from a specialized Macaca mulatta breeding company in Mexico City, called Proyecto Camina A.C., which follows international standards of reproduction and animal care. The company does not catch animals in the wild and has certified veterinary care. All the animal care, housing, and experimental procedures were approved in the protocol 0.27A by bioethics in Research Committee of the Instituto de Neurobiología, Universidad Nacional Autónoma de México The protocol follow the 3Rs and conformed to the principles outlined in the Guide for Care and Use of Laboratory Animals (NIH, publication number 85-23, revised 1985) and the NORMA Oficial Mexicana NOM-062-ZOO-1999, ‘Especificaciones técnicas para la producción, cuidado y uso de los animales de laboratorio’. The animals are housed in a monkey facility with cages of 2.2 m3, with controlled temperature, humidity and a 12-12 h day-night cycle. In our Institute we have three certified veterinarians that continuously provide care for the monkeys and perform regular health checkups and medical analysis of the animals. Animal care staff keep the facilities clean, and they provide food and water to the animals 365 days of the year. In addition, monkeys are monitored daily by researchers and the animal care staff to check their conditions of health and welfare. The animals are fed ad libitum with a special diet of the brand LabDiet (Monkey Diet 5038). In addition, they have daily access to fruits such as apples and bananas, as well as raisins and berries. The social intra-species enrichment includes monkey pairing in the same cage for compatible individuals, and cages that allow visual interaction. In addition, we play videos of Rhesus monkeys in the wild to the animals a couple of hours a day. Regarding the non-social enrichment, we routinely introduce toys (often containing food items that they liked) to their home cage to promote their exploratory behavior.

Stimuli and task

Humans. Participants performed a temporal interval categorization task (see Fig. 1A). In this task, participants had to categorize a visual stimulus based on its duration on screen. Each trial started with the presentation of a fixation cross for 2 s. Then, the to-be-categorized stimulus (a circle around the fixation cross) was shown for a specific time interval. After another 2-s delay, participants had to report whether the presented stimulus was “short” or “long” according to previously learned prototypes by pressing the right or left arrow key on a computer keyboard. Note that each block had a learning phase of ten trials in which participants would learn the meaning of “short” and “long” durations for that set (only the shortest and the longest intervals of each block were presented, allowing participants to implicitly learn the category boundary for that set). In order to avoid motor preparation, response mappings (i.e., left vs. right arrow key) were randomly changed on a trial-by-trial basis. Feedback was presented at the end of each trial via a colored fixation cross (green for correct and red for incorrect responses). The task involved three stimulus sets (T1, T2 and T3) with different interval durations (see Fig. 1B). This design allowed us to compare “short” and “long” decisions for stimuli with the same objective duration (see two black boxes in Fig. 1B). The different sets were presented in a blocked design, with order randomised per participant. A total of 336 trials (112 per block) were performed, with the experiment lasting for approximately 1 h. The mean accuracy was 69.10% (std = 11.01) for T1, 72.08% (std = 9.94) for T2 and 75.49% (std = 9.56) for T3.

Figure 1 Temporal categorization task.

(A) Schematic of the time-interval categorization tasks adopted for monkeys (first row) and humans (second row). Subjects had to indicate whether the test interval was of “long” or “short” duration. Response mapping was randomized per trial and shown after the decision delay. (B) Interval durations for each of the three sets. Note that certain intervals could be “short” in one set but “long” in another (marked with black outline). (C) Psychometric curves (probability of answering “long”) for humans (black) and one monkey (green), per stimulus set. Error bars represent standard error across subjects for humans and across sessions for the monkey. Shaded area depicts standard error from the mean (SEM).

Monkeys. Task details have been reported previously (Mendoza et al., 2018). In short, monkeys were trained to categorize the temporal interval between two visual stimuli as either “short” or “long”, according to previously learned prototypes. First, a circle containing a fixation point was shown in the center of the screen. Then, the animal started the trial by staring at the fixation point and by placing the cursor inside the central circle. After a variable waiting period (500–1,000 ms), two parallel bars separated by constant distance appeared for 50 ms, disappeared for a particular test interval, and reappeared in the same position. The first and second stimulus presentations indicated the beginning and the end of the test interval, respectively. After a fixed delay (500 ms) two response targets (orange and blue circles) were presented. Both response targets could occupy one of eight possible locations on the periphery of the screen. The monkeys were trained to move the cursor from the central circle to the orange target if the test interval was short or to the blue target if it was long. The monkey received a juice reward immediately after each correct response. The task involved three stimulus sets (T1, T2 and T3) with different interval durations (see Fig. 1B), presented in separate trial blocks. Each block had an initial instruction phase of 24 trials in which only the shortest and the longest intervals of each block were presented. In these trials the color of the parallel bars matched the color of the correct response target (orange for the short interval and blue for the long interval). The following 96 trials constituted the test phase in which the color of the bars was green regardless of the stimulus category. A total of 199 sessions (each of them involving 96 experimental trials) were performed. The mean accuracy was 68.59% (std = 5.02) for T1, 69.00% (std = 5.76) for T2 and 71.17% (std = 9.25) for T3.

Recordings

Humans. 96-electrode scalp EEG was collected using the BrainVision actiCAP system (Brain Products GmbH, Munich, Germany) with a sampling rate of 500 Hz. Electrodes were labelled according to the international 10-20 system. The reference electrode during the recording was Cz. Amplification and digitalization of the EEG signal was done through an actiCHamp DC amplifier (Brain Products GmbH, Munich, Germany) linked to BrainVision Recorder software (version 2.1; Brain Products GmbH, Munich, Germany). Vertical (VEOG) and horizontal (HEOG) eye movements were recorded by placing additional bipolar electrodes above and below the left eye (VEOG) and next to the left and right eye (HEOG).

Monkey. Neurophysiological recordings were performed as described in previous publications (Mendoza et al., 2018; Rassi et al., 2023). In short, recording chambers (8-mm inner diameter) were implanted over the left pre-SMA and dorsolateral prefrontal cortex (dlPFC) during aseptic surgery under Sevoflurane (1–2%) gas anesthesia. Chamber positions were determined on the basis of structural MRI. Titanium posts for head restraining were implanted on the skull. Broad spectrum antibiotics (Enrofloxacin, 5 mg/kg/day, i.m.) and analgesics (Ketorolac 0.75 mg/kg/6 h or Tramadol 50–100 mg/4–6 h, i.m.) were administered for 3 days after surgery. The extracellular activity of neurons in pre-SMA was recorded with quartz-insulated tungsten microelectrodes (1–3M Ω) mounted in multielectrode manipulators (Eckhorn System; Thomas Recording, GMbH, Giessen, Germany). All neurons were recorded regardless of their activity during the task, and the recording site changed from session to session. Spike waveform data were sorted online employing window discriminators (Blackrock Microsystems LLC, Salt Lake City, UT, USA). LFP data were simultaneously recorded from both pre-SMA and dlPFC using a 250-Hz low-pass filter and stored at 1,000 Hz for offline analysis. The titanium posts of the head-restraining implant were used for grounding.

Pre-processing of electrophysiological data

Humans. Pre-processing was performed in MATLAB R2021a using custom scripts and functions from EEGLAB (Delorme & Makeig, 2004) and Fieldtrip (Oostenveld et al., 2011) toolboxes. Data were first resampled to 250 Hz and filtered between 0.5 and 30 Hz. Noisy electrodes were automatically detected (EEGLAB function clean_channels) and interpolated. EEG data were re-referenced to the common average and independent component analysis (runica algorithm) was performed. An automatic component rejection algorithm (IClabel) was employed to discard components associated with muscle activity, eye movements, heart activity or channel noise (threshold = 0.8) (Pion-Tonachini, Kreutz-Delgado & Makeig, 2019). In addition, components with an absolute correlation with HEOG, VEOG or ECG channels higher than 0.8 were discarded. Furthermore, artifact subspace reconstruction (ASR) was employed to correct for abrupt noise with a cut-off value of 20 SD (Chang et al., 2019) ERPs were obtained by averaging trials within subjects, condition and electrodes. In order to reduce the dimensionality of the data, Principal Component Analysis (PCA) was used to compute spatial filters that explained most of the variance of the EEG data(Guarnieri et al., 2020; Zanotelli, Filho & Tierra-Criollo, 2010). We concatenated ERPs across subjects to compute common spatial filters for all subjects (i.e., Group PCA analysis) (Dien, 2012).

Monkeys. All LFP pre-processing was done with Fieldtrip and custom MATLAB R2019a code. Only data from pre-SMA was used based on prior work (Kononowicz & Penney, 2016). Epochs were visually inspected and excessively noisy channels and trials were rejected (around 10% of data) (Rassi et al., 2023). Data were filtered between 0.5 and 30 Hz. ERPs were obtained by averaging trials within pre-SMA electrodes, sessions and conditions.

Statistical analysis

For the behavioural analysis, we calculated psychometric curves per subject (in humans) or per session (in monkeys). For this purpose, the probability of categorizing each interval of the corresponding set of stimuli as “long” was fitted with a logistic function, which was defined as: fx=11+e−ax−b.

The parameter a represents the steepness of the slope and b is the sigmoid midpoint. The slope was extracted per subject (in humans) or per session (in monkeys) to be statistically compared between sets and species. These comparisons were done using ANOVAs and t-tests (paired and independent) as implemented in MATLAB.

For the electrophysiological data, a cluster-based permutation test (Maris & Oostenveld, 2007) was used to assess condition-related statistical differences in ERPs. In short, this test uses non-parametric Montecarlo randomization in order to control for the type I error rate arising from multiple comparisons while taking into account the dependency of the data (see Maris & Oostenveld, 2007). The significance level for the cluster permutation test was set to 0.025 (corresponding to a false alarm rate of 0.05 in a two-sided test). A paired-samples t-test was chosen as the first-level statistic to compare experimental conditions in humans and monkeys.

For the comparison of monkey and human ERPs we performed a cross-correlation analysis. In short, the Pearson correlation coefficient was computed between the average human ERP (per electrode and across subjects) and the monkey ERP across conditions for different time lags. Because the lengths of the human and monkey post-interval ERPs were different (i.e., the delay was longer in humans; see Fig. 1A), the human ERP time series was resampled to match the monkey ERP in length. The false discovery rate method was adopted to correct for multiple comparisons (Benjamini & Hochberg, 1995).

Results

Behaviour

The psychometric curves of both the human participants and the monkey followed a typical sigmoid shape showing that the probability of categorizing a particular interval as ‘’long” increased as a function of the interval duration (see Fig. 1C).

The slope of the psychometric curves, which reflects sensitivity, differed significantly between sets in both the monkey (F (2,134) = 51.93; p < 0.001) and humans (F (2,69) = 5.54; p = 0.006). In line with previous literature (Mendez et al., 2011), post-hoc t-tests showed that the slope of the psychometric curve became flatter in blocks with longer stimulus duration, thereby reflecting decreased sensitivity with longer intervals in both species. Specifically, post-hoc t-tests showed that in humans the slope of the psychometric curve was significantly flatter in set T2 relative to set T1 (t(23) = 3.30; p = 0.003) and in set T3 relative to set T1 (t(23) = 3.27; p = 0.003), but not in set T3 relative to set T2 (t(23) = 0.93; p = 0.35). In the monkey, the slope of the psychometric curve was significantly flatter in set T3 relative to both set T1 (t(86) = 8.47; p < 0.001) and set T2 (t(83) = 3.43; p < 0.001), and in set T2 relative to set T1 (t(99) = 7.07; p < 0.001). When directly comparing the slope of the psychometric curve between species, we found that humans had greater sensitivity (i.e., steeper slope) than monkeys in set T1 (t(74) = 2.64; p = 0.01), set T2 (t(71) = 3.67; p < 0.001) and set T3 (t(59) = 2.39; p = 0.020).

In sum, these behavioural results indicate that both humans and the monkey successfully categorized time intervals and that the psychometric properties of behavioural responses were similar across species.

Post-interval evoked responses in humans

We first sought to replicate and extend previous findings showing differences between “long” and “short” temporal decisions in post-interval ERPs of human EEG. In order to reduce the dimensionality of the data, we performed a group PCA. We selected the first three components for further analysis since they cumulatively explained over 70% of the variance (40.3%, 24.9% and 11.7%, respectively). Only the first two components showed significant differences between “short” and “long” decisions (see below).

The first principal component (Fig. 2A) showed a more pronounced positive potential around 300 ms for correct “short” decisions (tcluster = −113.89; pcluster = 0.002) and a more pronounced negative potential around 600 ms for correct “long” decisions (tcluster = −100.59; pcluster = 0.002; Fig. 2B). The same pattern of results was observed when selecting correct trials with the same objective duration (i.e., matched for physical stimulus properties but belonging to a different category), although in this case, only the difference around 300 ms remained significant (tcluster = −95.64; pcluster = 0.003; Fig. 2C). No significant differences were found for incorrect trials (Fig. 2D).

Figure 2 Post-interval evoked potentials in humans.

Each row depicts the ERPs after a Group PCA-derived spatial filter is applied. (A) Topography of the spatial filters for the first principal component. (B) ERPs for “long” and “short” responses for correct trials. (C) ERPs for “long” and “short” responses for correct trials with the same objective (but differently perceived) duration. (D) ERPs for “long” and “short” responses for correct trials. (E–H) same as A–D for second component. Statistical significance at p < 0.025 is marked with a black line, and 0 on the x-axis represents the offset of the stimulus whose duration was to be evaluated. Shaded area depicts standard error from the mean (SEM).

The second principal component (Fig. 2E) showed a more pronounced positive potential around 200 ms for “long” decisions (tcluster = 231.63; pcluster < 0.001), a more pronounced positive potential around 400 ms for “short” decisions (tcluster = −283.94; pcluster < 0.001), and a more pronounced negative potential around 1,000 ms for “short” decisions (tcluster = 64.89; pcluster = 0.024; Fig. 2F). The same pattern of results was observed when selecting correct trials with the same objective duration (tcluster = 189.78, pcluster = 0.002; tcluster = −99.44, pcluster = 0.003; tcluster = −71,03, pcluster = 0.009; Fig. 2G). Crucially, this pattern of results was reversed for incorrect trials for the ERPs around 200 ms and 400 ms (tcluster = 49.91, pcluster = 0.009 and tcluster = −45.96, pcluster = 0.018, respectively; Fig. 2H).

Based on previous literature, we repeated the analysis using a baseline correction of 200 ms prior to the offset of the stimulus to attenuate the putative effects of evoked potentials emerging during the presentation of the visual stimulus (Kononowicz & Rijn, 2014). As depicted in Fig. S2, we obtained a qualitatively similar pattern of results.

Together, these results show that post-interval potentials reflect the perceived duration of the stimuli as (i) ERPs for “short” and “long” decisions differed significantly even if the objective duration of the stimulus was the same, and (ii) some of these ERP effects were reversed for incorrect trials.

Post-interval responses in human and non-human primates

In order to compare post-interval potential differences between “long” and “short” temporal decisions across species, we selected a group of frontal electrodes in humans that overlaps spatially with the recording locations in the monkey (Fig. 3AE).

Figure 3 Fronto-central post-interval evoked responses in human and non-human primates.

(A) Topography showing the location of electrode selection in humans. (B) ERPs for “short” (orange) and “long” responses (blue) in humans for correct trials. (C) ERPs for “short” (orange) and “long” responses (blue) in humans for correct trials with the same objective (but differently perceived) duration. (D) ERPs for “short” (orange) and “long” responses (blue) in humans for incorrect trials. (E) Analysed recording sites in the monkey (pre-SMA). (F–H) Same as B–D but for the monkey. (I–K) Same as B-D but depicting monkey’s firing rate instead of ERPs. Statistical significance at p < 0.025 is marked with a black line in each subplot, and 0 on the x-axis represents the offset of the stimulus whose duration was to be evaluated. Shaded area depicts standard error from the mean (SEM).

In humans, correct “short” decisions were associated with a more pronounced negative potential in the first 100 ms (tcluster = 123.64; pcluster = 0.002), a more pronounced positive potential around 200 ms (tcluster = −166.69; pcluster < 0.001) and a more pronounced negative potential around 1000 ms (tcluster = 56.42; pcluster = 0.021; Fig. 3B). A qualitatively similar pattern of results was observed when selecting trials with the same objective (but differently perceived) duration (tcluster = 68.53; pcluster = 0.016; tcluster = −108.58; pcluster = 0.003; Fig. 3C). No significant differences were identified for incorrect trials (Fig. 3D).

Similarly, correct “short” decisions in the monkey were associated with a more pronounced negative potential in the first 100 ms (tcluster =13475; pcluster < 0.0001), a more pronounced positive potential around 200 ms (tcluster = −71.18; pcluster = 0.004) and a more pronounced negative potential around 300 ms (tcluster = 120.28; pcluster = 0.004; Fig. 3F). A qualitatively similar pattern of results was observed when selecting trials with the same objective (but differently perceived) duration (tcluster = 1147; pcluster < 0.001; Fig. 3G). Moreover, this pattern of results was significantly reversed for incorrect trials (tcluster = −71.96; pcluster < 0.001; tcluster = −333.35; pcluster < 0.001; Fig. 3H). In addition, relative firing rate in the monkey was more pronounced for correct “short” decisions around 300 ms for all trials (tcluster = −197.68; pcluster = 0.002; Fig. 3I) as well as for trials matched for duration (tcluster = −139.48; pcluster = 0.004; Fig. 3J), and this pattern of results was reversed for incorrect trials (tcluster = 252.63; pcluster < 0.001; Fig. 3K).

Note that a qualitatively similar pattern of results was obtained when using a baseline correction of 200 ms prior to the offset of the stimulus (see Fig. S3).

In order to quantify the qualitative similarities between the monkey and human ERPs, we performed a cross-correlation analysis. Specifically, we computed the Pearson correlation coefficient between the average human and monkey ERPs across conditions for different time lags. This analysis revealed that human and monkey ERPs were maximally correlated for a lag of 31 samples (maximal r-value = 0.83; see Fig. 4AB). For this lag, the majority of the electrodes showed significant correlations after correction for multiple comparisons (pfdr < 0.001). Correlations were positive for frontocentral electrodes and negative for occipital electrodes (Fig. 4A).

Figure 4 Cross-correlation between human and monkey ERPs.

(A) Topography depicting the r-values of the correlation between the average human and monkey post-interval potential (lag of 31 samples). (B) Depiction of the human and monkey ERPs. The human ERP was obtained by computing a spatial filter with the previously obtained r-values.

Together, these results show that: (i) post-interval potentials in monkey pre-SMA and human frontocentral EEG are similar and reflect perceived time durations, and (ii) some of these differences are also mirrored in firing rates modulations in the monkey.

Discussion

In this study, we assessed whether monkeys and humans share the same neural mechanisms for time estimation. For this purpose, we analysed the electrophysiological signals of 24 humans (EEG) and one monkey (extracellular recordings in pre-SMA) while they categorised a temporal interval as either “short” or “long” based on previously learned prototypes. Our results show that evoked potentials after the presentation of the time interval differed significantly between “short” and “long” decisions in both humans and monkeys. Crucially, we show that these differences reflect the perceived (and not the objective) duration of the time intervals because: (i) the same difference in post-interval potentials was evident when stimuli had the same objective (but differently perceived) duration, and (ii) the reversed pattern of results was observed for incorrect trials. In addition, some of the differences in post-interval potentials were accompanied by significant changes in monkey’s firing rates.

Previous literature has shown significant differences in post-interval ERPs as a function of the perceived duration in human EEG (Damsma, Schlichting & Van Rijn, 2021; Kononowicz & Rijn, 2014; Kruijne, Olivers & Van Rijn, 2021; Lindbergh & Kieffaber, 2013; Özoğlu & Thomaschke, 2023; Tarantino et al., 2010). These potentials are thought to reflect comparison and decision processes (Kononowicz & Rijn, 2014; Lindbergh & Kieffaber, 2013). Our results replicate these findings in humans and show that these potentials are encompassed in two different components (P300 in the first PCA and P200/LPP in the second PCA), which suggests different neural generators (Dien, 2012). In addition to previously reported post-interval potentials, we found significant differences between “short” and “long” decisions in a slow negative potential (occurring after 500 ms). This slow negative potential peaked around 600 ms for “long” perceived durations and around 1,000 ms for “short” perceived durations (Fig. 2B). Based on previous literature, we speculate that these slow frontocentral negative potentials reflect working-memory processes. Working memory retention has been associated with slow cortical potentials lasting from 200 ms to several seconds that seem to vary depending on the type of stimulus and cognitive load (Bosch, Mecklinger & Friederici, 2001; Ruchkin et al., 1992; Schneider et al., 2020). Because in the temporal bisection task the duration of the presented temporal interval has to be kept in working memory during the delay before a motor response is performed (Treisman, 2013), ERPs associated with memory retention are expected during this period.

It has been proposed that the differences in post-interval potentials as a function of perceived duration observed in humans reflect differences in the timing of cognitive processes supporting time estimation, rather than a timing mechanism in itself (Kononowicz & Penney, 2016; Kononowicz & Rijn, 2014; Lindbergh & Kieffaber, 2013). It can be argued that when the interval is longer than the category boundary, subjects can make a decision before the interval offset. For stimuli shorter than the category boundary, participants are only able to decide after the interval offset (Mendoza et al., 2018). This interpretation is supported by differences in the ERPs reported here. Specifically, comparison/decision processes could be reflected in the more pronounced P200 for long decisions and in the later P300/LPP for short decisions (Fig. 2BF). Memory retention of the decision could be reflected in the negative slow potential peaking at 600 ms for long decisions and at 1,000 ms for short decisions.

In order to compare human and monkey post-interval neural responses, we selected a cluster of frontocentral electrodes in humans that overlapped with the location of the recordings in the monkey (around pre-SMA). Strikingly, the neural dynamics observed in both species were highly similar, showing significant differences between short and long perceived durations with the same polarity. In addition, the changes in the later post-interval potential in the monkey were mirrored in the firing rate, which suggests differential excitation levels in pre-SMA for short and long perceived durations. Since we show shared neural substrate of time categorization between humans and monkeys, our findings support the idea that research on the monkey brain can help elucidate the neural mechanisms supporting human time estimation, and, by extension, its related deficits in clinical populations (Merchant et al., 2008).

The main limitations of the current work are related to differences between species in terms of experimental design and recording methods used. First, the temporal categorization task was not exactly the same in the human and monkey experiments. The tasks differed in both the presentation of the temporal interval (empty interval in monkey vs. filled interval in human) and the duration of the post-interval delay (2 s in human and 0.5 s in monkey). Recordings in humans were done with scalp EEG, while recordings in monkeys were with intracortical electrodes. These factors may have affected the shape of the ERPs and should be controlled for in future research. To avoid possible confounders, future studies should: (i) include EEG recordings in NHP, and (ii) make sure that both species perform exactly the same task.

In conclusion, this study extends previous findings regarding post-interval evoked potentials in the context of time estimation in humans and shows that similar neural mechanisms are present in monkeys. Therefore, our results further support the idea that the monkey brain is a good model to investigate the neural mechanisms underlying human time perception.

Supplemental Information

Supplemental Information 1 Supplementary Figures

Supplemental Information 2 The ARRIVE guidelines 2.0: author checklist

Additional Information and Declarations

Competing Interests

Author Contributions

Human Ethics

Animal Ethics

Data Availability

The authors declare there are no competing interests.

Julio Rodriguez-Larios conceived and designed the experiments, performed the experiments, analyzed the data, prepared figures and/or tables, authored or reviewed drafts of the article, and approved the final draft.

Elie Rassi analyzed the data, authored or reviewed drafts of the article, and approved the final draft.

German Mendoza conceived and designed the experiments, performed the experiments, authored or reviewed drafts of the article, and approved the final draft.

Hugo Merchant conceived and designed the experiments, authored or reviewed drafts of the article, and approved the final draft.

Saskia Haegens conceived and designed the experiments, authored or reviewed drafts of the article, and approved the final draft.

The following information was supplied relating to ethical approvals (i.e., approving body and any reference numbers):

Institutional Review Board (IRB) of the New York State Psychiatric Institute

National University of Mexico Institutional Animal Care and Use Committee

The following information was supplied relating to ethical approvals (i.e., approving body and any reference numbers):

National University of Mexico Institutional Animal Care and Use Committee and conformed to the principles outlined in the Guide for Care and Use of Laboratory Animals (NIH, publication number 85–23, revised 1985).

The following information was supplied regarding data availability:

The data and code are available at OSF: Rodriguez-Larios, Julio. 2024. “Common Neural Mechanisms Supporting Time Judgements in Humans and Monkeys.” OSF. June 14. https://osf.io/tm9bz/.

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
