# Peer review of "Common neural mechanisms supporting time judgements in humans and monkeys"

_PeerJ, doi:10.7717/peerj.18477_

## Round 0.1 · original submission · Major Revisions

Your manuscript has now been seen by two reviewers. You will see from their comments below that while they find your work of interest, some major points are raised. We are interested in the possibility of publishing your study, but would like to consider your response to these concerns in the form of a revised manuscript before we make a final decision on publication. We therefore invite you to revise and resubmit your manuscript, taking into account the points raised. Please highlight all changes in the manuscript text file.

Reviewer 1 ·

Basic reporting

Article is clearly written, sufficiently and appropriately referenced, conforms in format with relevant and clear figures, and constitutes a unit of publication.

Experimental design

Article describes original primary research with well-defined and relevant question. Analyses are carried out to a sufficient standard with clear description of methods.

Validity of the findings

Findings are valid, data is provided, statistics are appropriate, conclusions are clearly stated and appropriately drawn from the results.

Additional comments

My main comment is that if it is possible for the number of NHP subjects used in the study to be increased, that would be helpful, since the current number of subjects is one. As far as I understood, the NHP data is drawn from Mendoza et al. 2018 which had two subjects?

Beyond this, if it is possible for the authors to demonstrate in more detail the degree to which human and nonhuman electrophysiological responses are similar, that would strengthen the paper, given that currently the authors only describe a qualitative similarity.

Reviewer 2 ·

Basic reporting

- This is a well-written and structured study. The authors have conducted research on a significant question for timing researchers – the neural underpinnings of time perception. Recent studies have shown that post-interval potentials reflect subjective time. The current study provides an animal-human comparison using a temporal categorization task, which is relevant to previous findings. The authors hypothesize that short-long perceptions are reflected in post-interval potentials and firing rates. They have found significant differences between short and long responses in both human and non-human primate (NHP) data. Overall, their research question falls within the journal’s scope. The figures are well-made, and the language is clear. The authors have shared the data and relevant scripts. However, I have several points to address before suggesting this article for publication.
- The authors have cited important and relevant publications in the introduction. However, the way these publications are included is limited to the statement (except citations) “However, it has been recently shown that time estimation is better reflected in post-interval potentials. In this way, it has been shown that post-interval potentials in fronto-central electrodes differ significantly depending on the perceived time duration“. This section should be more informative regarding the potentials, tasks, and main findings. I suggest including these studies in a more detailed way.

Experimental design

- I find the description of the task a little confusing. Authors wrote “Each block had a learning phase of ten trials in which participants would learn the meaning of short and long durations for that set (only the shortest and the longest intervals of each block were presented. This allowed participants to implicitly learn the category boundary”. They also wrote “After another 2 seconds delay, participants had to report whether the presented stimulus was short or long by pressing the right or left arrow key on a computer board.” What do response “short” and “long” refer to in this context? Does the participant report if the presented interval is identical to the short or long durations of that set? Based on the psychometric functions I assume the question is rather whether time intervals are closer to short or long durations. If not, then what response did participants choose for the intervals that are spaced between short and long anchors? And how many times were the intervals presented?
- Comparable timing studies on the same question (which are also cited in this manuscript) did not use trial-wise feedback. What was the authors’ reasoning for choosing to use feedback?

Validity of the findings

- In the preprocessing stage of human EEG data analysis, have you used a baseline correction concerning the built-up EEG signal that occurs before the interval offset? The ramping negative activity (also studied as Contingent Negative Variation or CNV) usually found in timing studies and observed during the presentation of the time stimulus is likely to affect the potentials measured immediately after the offset (Kononowicz, T. W., & van Rijn, H. (2014). Decoupling interval timing and climbing neural activity: a dissociation between CNV and N1P2 amplitudes. Journal of Neuroscience, 34(8), 2931-2939.). Studies you cited have used baseline correction and a 1–20 Hz filter to minimize contamination. If you have not controlled for this effect, could you check if your findings are similar after doing so?
- The authors describe behavioral data as being similar across species, with human data showing significantly more sensitivity. Considering the trial-wise feedback, this is expected. I would like to ask about the overall performance and how it affected EEG analysis. In Figure 3, we see ERPs for correct and incorrect trials. How many incorrect trials were observed in the end? Especially for the time interval set 3, the success rate seems quite high. Were correct and incorrect trials comparable?

Additional comments

1. The authors found a slow negative peak around 1000 ms post-interval, which they associate with working memory retention. I would like more clarification on this explanation. Does this refer to the updated temporal representation for the short and long durations after the previous trial’s feedback, or the temporal representation regarding the current presented interval?
2. Despite the challenges of designing a study suitable for both animals and humans, the authors conducted a well-structured study. However, there are some minor differences that could have been equated but were not. For example, why is the fixation period at the beginning of the trial fixed (2 s) for humans and variable (0.5–1 s) for monkeys?
3. Please read the manuscript thoroughly for spelling errors such as those in lines 120 and 148.

---

## Round 0.2 · accepted · Accept

Thank you for the revised manuscript and the detailed response letter. We are delighted to accept your manuscript for publication.

Reviewer 2 ·

Basic reporting

No comment

Experimental design

No comment

Validity of the findings

No comment

Additional comments

No comment